# Lavage Volume of Arthrocentesis in the Management of Temporomandibular Disorders: A Systematic Review and Meta-Analysis

**DOI:** 10.3390/diagnostics12112622

**Published:** 2022-10-28

**Authors:** Hei Christopher Tsui, Chun Mo Lam, Yiu Yan Leung, Kar Yan Li, Natalie Sui Miu Wong, Dion Tik Shun Li

**Affiliations:** 1Faculty of Dentistry, The University of Hong Kong, Hong Kong, China; 2Oral and Maxillofacial Surgery, Faculty of Dentistry, The University of Hong Kong, Hong Kong, China; 3Clinical Research Centre, Faculty of Dentistry, The University of Hong Kong, Hong Kong, China

**Keywords:** arthrocentesis, temporomandibular joint disorders, temporomandibular joint, therapeutic irrigation

## Abstract

The aim of this study was to investigate the most effective lavage volume of arthrocentesis in the management of temporomandibular disorders. A comprehensive electronic search, based on the PRISMA guidelines, was performed, which included a computer search with specific keywords, a reference list search and a manual search. The inclusion criteria were the following: a randomized controlled trial, at least 20 subjects who underwent arthrocentesis, mention of the irrigation materials used for the arthrocentesis, mention of the irrigation volumes used for the arthrocentesis, MMO and pain measured as VAS or NRS, were reported as outcome figures, mention of a specific diagnosis or signs and symptoms, and inclusion of the data on the MMO or VAS/NRS at 6-month follow-up. Sixteen publications were enrolled in the meta-analysis, comparing arthrocentesis with a lavage volume <150 mL and arthrocentesis with a lavage volume ≥150 mL, in the efficacy of the improvement in the mouth opening and pain reduction. The results revealed the group with a lavage volume <150 mL had a greater improvement in the mouth opening and pain reduction. However, results are to be interpreted with caution, due to the paucity of the randomized controlled literature and other confounding factors. Further high-quality studies are required to provide a better conclusion to the treatment outcomes of the different lavage volumes.

## 1. Introduction

Temporomandibular disorders (TMDs) are a series of clinical problems which affect the temporomandibular joint (TMJ), masticatory muscles and associated structures [1]. Based on the diagnostic criteria for a TMD (DC/TMD), established in 2014, a TMD was diagnosed, based on the physical examination (Axis I) and assessment of the psychosocial status and pain-related disability (Axis II) [1]. In more detail, the DC/TMD Axis I includes: (a) muscle disorders, including myalgia, myofascial pain and myofascial pain with referral, etc., (b) intra-articular joint disorders, including disc displacement with or without the reduction or mouth opening limitation; (c) other articular conditions, including arthralgia.

TMDs are the second most common musculoskeletal problem following chronic lower back pain. Around 6–12% of the general population is thought to be affected by TMDs [2,3]. TMDs usually affect people between 20 to 40 years of age, and are more prevalent in females [4,5]. Some of the common signs and symptoms are facial pain, limited mouth opening and joint sounds. In the United States, the estimated management cost was about USD 4 billion per year [6]. Moreover, patients with TMDs used a broader range of services and hence, consumed more resources [7].

The management of TMDs focuses on alleviating pain or joint noises, restoring normal joint function and improving the overall quality of life. The first line approach involves the non-surgical treatment that includes a soft diet, pharmacotherapy, such as non-steroidal anti-inflammatory drugs (NSAIDs), occlusal splint therapy and physiotherapy [8]. In particular to the muscle-related TMD patients, the conservative approaches, such as physical therapy, laser therapy, occlusal splints and acupuncture were effective in pain reduction [9]. Other than the conservative modalities, pharmacologic agents are widely used for the treatment of mild and moderate TMD. Common drugs that improved TMD pain include NSAIDs, opioids, corticosteroids, antidepressants, anticonvulsants, antiepileptics, muscle relaxants, sedatives and hypnotics [10]. The intramuscular injections of botulinum toxin (BTX) had shown to relieve the muscle pain from TMDs. Intra-articular joint injections with corticosteroids, hyaluronic acid (HA) and platelet-rich plasma, also showed improvement in pain and functions [10]. Novel agents, such as ozone, were used as a topical gaseous therapy to the muscles and injective agents at the temporomandibular joint and demonstrated promising results in the muscle and the articular TMD [11]. 

Although the success rate of non-surgical treatments is approximately 70%, some patients do not respond well to these treatments. Patients who are refractory to non-surgical therapies and have high levels of pain and dysfunction, are suitable for surgical interventions, such as meniscectomy, disc repositioning and condylotomy [12]. These invasive procedures are often associated with surgical risks. Minimal invasive procedures, such as arthrocentesis, serve as an appropriate alternative to surgical intervention [13]. Indications of arthrocentesis, included patients with internal TMDs not responding to conservative treatment, patients with anterior disc displacement with or without reduction, disc adhesions, synovitis/capsulitis and degenerative osteoarthritis. A review article concluded that arthrocentesis is a highly effective approach when taking into account the notable clinical benefits and the small number of complications [13,14].

Arthrocentesis of the TMJ was first described by Nitzan and applied on patients with severe, limited mouth opening [15]. Arthrocentesis of the TMJ refers to the lavage of the upper joint space with saline, without visualizing the joint. Studies showed that arthrocentesis decreased pain, increased the maximal incisal opening and the follow-up showed the prolonged relief of symptoms [16]. This treatment utilized the pumping actions and hydraulic pressure to remove adhesions and inflammatory mediators, and widened the joint space [17,18]. Studies suggested that arthrocentesis reduced pain and the functional impairment rapidly, as an initial therapy, when compared to conventional treatment [19]. With a success rate of over 80% [20], and being less invasive than surgical interventions, arthrocentesis has become a common therapeutic intervention for patients with TMDs. 

Recent studies have investigated the factors that determine the effectiveness of arthrocentesis, such as the needle technique, adjunctive treatment and lavage volume [21,22,23]. In particular, there is no consensus on the lavage volume in arthrocentesis, and the commonly adopted volume ranges from 50 mL to 300 mL. When arthrocentesis was introduced in the TMJ, approximately 200 mL of lactated Ringer’s solution was used [15]. Studies have suggested that a smaller lavage volume was equally effective in washing the upper joint space of the TMJ [22,24]. One study suggested that a change in lavage volumes did not provide a statistical significance on the reduction of pain and the maximum mouth opening [22]. To the best of the authors’ knowledge, no meta-analysis has compared the different lavage volumes on the effectiveness of arthrocentesis, in terms of the pain level and the maximum mouth opening. The aim of this study is to determine whether different lavage volumes will affect the treatment outcome of arthrocentesis, in relation to pain and mouth opening. 

## 2. Materials and Methods

### 2.1. Protocol and Registration

The PRISMA 2020 statement [25] was taken as the reference in reporting this systematic review and meta-analysis, while this review was not registered in PROSPERO.

### 2.2. Eligibility Criteria

Criteria for selection was based on the PICOTS framework, as follows: ‘P’ (population): adult humans with a definitive clinical diagnosis or specific signs and symptoms of TMDs. ‘I’ (intervention): arthrocentesis, or lysis and lavage, with a clear indication of irrigation volumes and materials used. ‘C’ (comparison): control or adjunctive treatment including an occlusal splint or physiotherapy, or compared with arthrocentesis with intra-articular injections, or compared with arthrocentesis with different intra-articular injections, or compared with arthrocentesis on patients with different diagnoses of TMDs. ‘O’ (outcomes): the primary outcome is un-assisted/undefined, painless maximum mouth opening (MMO) in millimeters. The secondary outcome is pain intensity at rest, measured by the visual analogue scale (VAS) or the numerical rating scale (NRS). ‘T’ (time): all studies should have their follow-up period of at least 6 months. ‘S’ (study design): a randomized, controlled clinical trial.

The exclusion criteria were as follows: studies not in English, full text not available, non-adult human studies, studies not related to TMDs or where there was no mention of specific diagnoses or signs and symptoms, studies not using arthrocentesis as intervention, studies that included less than 20 patients undergoing arthrocentesis, studies not using the MMO and VAS as measuring outcomes, non-clinical studies, technical notes, case reports and case series.

### 2.3. Data Collection and Processing Strategy

The search was conducted in a total of three rounds. In the first round, an electronic search in PubMed, Cochrane Library, EMBASE (OVID), Scopus and Web of Science was performed, updated to December 5, 2020. The following search terms were used: (“Temporomandibular Joint” OR “Temporomandibular Joint Disorders” OR “Temporomandibular Joint Dysfunction Syndrome” OR “Craniomandibular Disorders” OR TMJ OR TMD or CMD) AND (Arthrocentesis OR “Temporomandibular Joint Arthrocentesis” OR lysis OR lavage). The publication date, language or publication status were not restricted. The articles obtained from the search term were imported into EndNote 20 and duplicates were removed with the software. The abstracts of the articles were then reviewed for eligibility. The full texts of the eligible studies after the first-round screening were obtained and imported into EndNote 20 and were included in the second-round screening.

In the second round, a manual search of the oral and maxillofacial surgery-related journals was performed in three relevant international journals: International Journal of Oral and Maxillofacial Surgery, the Journal of Oral and Maxillofacial Surgery, and the Journal of Cranio-Maxillofacial Surgery. Moreover, the reference lists of all identified studies from the first round and from the manual search were also scanned for relevant articles relating to the management of TMDs with arthrocentesis. The relevant articles after the first and second rounds of screening were included for the third-round evaluation.

In the third-round screening, the full texts of the included studies were evaluated, based on the following inclusion criteria: (1) randomized controlled trial; (2) at least 20 subjects who underwent arthrocentesis; (3) mention of the irrigation materials used for arthrocentesis; (4) mention of the irrigation volumes used for arthrocentesis; (5) the MMO and pain measured as VAS or NRS were reported as outcome figures; (6) mention of the specific diagnosis or signs and symptoms; (7) inclusion of the data on the MMO or VAS/NRS at 6-month follow-up. Articles after the third round of screening were considered eligible for inclusion for critical appraisal. A standard form was devised for the evaluation of inclusion/exclusion criteria of the studies screened.

### 2.4. Data Extraction Strategy

Data from the eligible studies were extracted using a standard data extraction sheet, specifically designed for this review. The data items that were extracted and analyzed were as follows: sample size, age at treatment, diagnosis of a TMD, change in the MMO in millimeters, change in pain measurement, time of follow-up, adjunctive procedure with arthrocentesis, such as an occlusal splint, intra-articular injections, irrigation materials and irrigation volumes.

### 2.5. Risk of Bias Analysis

With the revised Cochrane risk-of-bias tool (RoB 2) [26], two authors (H.C.T. and N.S.N.W.) critically appraised the eligible studies from the third round, for the risk of bias analysis, for verifying their strength in scientific evidence. Five domains were set for the appraisal, as follows: (1) randomization process; (2) deviations from the intended interventions; (3) missing outcome data; (4) measurement of the outcome; (5) selection of the reported result. When there was any discrepancy during the appraisal process between the two reviewers, the modulation was performed by the third reviewer (D.T.S.L.).

### 2.6. Summary Measures

For the continuous data, the weighted mean difference (WMD) was used to calculate the MMO (in millimeters) and pain (VAS from 0–10). Due to the various different follow-up time points in the different studies, the values at the 6-month follow-up or equivalent (i.e., 6 months, 24 or 26 weeks, or 180 days) were used for the meta-analysis. The VAS scales were standardized to a scale of 0–10. To obtain the mean and standard deviation (SD) in the studies where the range and median were given, the statistical formulas 5 and 16, in the article by Hozo et al. were used [27]. The postoperative SD values were assumed to be the same as the preoperative values when only the preoperative MMO and VAS were available. The values were excluded from the meta-analysis when the SDs of both the preoperative and postoperative values were not available.

### 2.7. Data Synthesis and Analysis

The extracted WMDs of the continuous data (MMO and VAS) were used in the meta-analysis. Forest plots were constructed using the random effects model with a 95% confidence interval (CI). Heterogeneity between the studies was evaluated using the Chi^2^ with *p* < 0.10 or I^2^ statistic of >50% [28]. In order to access whether the lavage volume had any effect on the clinical outcomes, we divided the included studies into two groups: (1) arthrocentesis with a lavage volume <150 mL; and (2) arthrocentesis with a lavage volume ≥150 mL. The MMO and VAS of the two groups were compared. The STATA (StataCorp. 2019. Stata Statistical Software: Release 16.0. College Station, TX, USA) was used for all statistical analyses.

### 2.8. Risk of Bias across Studies

Funnel plots were used to measure the publication bias, defined as the tendency to publish the results that are statistically or clinically significant. This method is deemed suitable, with more than 10 studies in the meta-analysis [28]. 

## 3. Results

### 3.1. Study Selection 

PRISMA 2020 flow diagram of the study is presented in Figure 1. An electronic database search resulted in a total of 2648 articles. Then, 645 articles remained after the removal of the duplicates. Following an initial screening of the titles and abstracts, 420 articles were excluded, due to the irrelevant topics. A total of 225 articles were included in the second round search. In the second round search, a manual search from 2010 to 2020 and a reference list search from the included studies resulted in five additional articles. Two hundred and thirty articles were included in the third-round evaluation. Of those, 214 articles were excluded due to the failure of one or more of the inclusion criteria, mentioned above. Thus, a total of 16 studies were included in the meta-analysis. 

### 3.2. Study Characteristics 

Details of the included studies are shown in Table 1. All studies are randomized control trials in the critical appraisal and meta-analysis. A total of 677 patients received arthrocentesis treatment for TMDs in the 16 included studies. In the included studies, the diagnoses included disc displacement with or without reduction (DDWR/DDWOR), Wilkes stages 3 and 4, internal derangement, osteoarthritis and arthralgia. In five of the included studies, less than 150 mL of lavage volume was used while eleven studies used more than or equal to 150 mL. Regarding the operative technique, 12 studies performed the traditional single-needle puncture while four other studies employed the double-needle technique. The intra-articular irrigants were saline and lactated Ringer’s solution. Some studies adopted the adjunctive injection of hyaluronic acid (HA), dexamethasone, platelet-rich plasma (PRP) and bone marrow nucleated cells. The follow-up period in the included studies, ranged from immediately after the procedure to 24 months. For this review, the outcome data for 6 months, 24 or 26 weeks, or 180 days of follow-up were extracted.

### 3.3. Risk of Bias within the Studies 

The assessment of the quality of the studies was carried out. Twelve studies showed a low risk of bias, three studies showed some concerns and one study showed a high risk of bias (Table 2). 

### 3.4. Synthesis of the Results 

The WMD and SD of the continuous variables were used for the meta-analyses. In one of the studies where the range and median were given [32], the mean and SD were estimated with statistical formulae [27]. In one study [34], the SD was calculated from the confidence intervals [28]. In two studies, the SD for the preoperative and postoperative pain measurements (VAS) were not available [29,30], therefore, the pain measurements from those studies were excluded from the meta-analysis. In three studies, the SD values of the postoperative MMO and VAS were not available. The SD of the postoperative MMO and VAS were assumed to be the same as the pre-operative values. 

Random effects model was used for the construction of the forest plots, due to the intention of the generalization inference and the substantial heterogeneity found in the included studies (I^2^ ranged from 70.02% to 98.5%; all *p* < 0.1). 

### 3.5. MMO 

The forest plots of the pooled WMD, between the <150 mL and ≥150 mL groups, in the improvement of the MMO, are summarized in Figure 2. All included studies demonstrated an improvement in the MMO after arthrocentesis (range 1.85–16.14 mm), with a greater improvement in the MMO in the group with a lavage volume <150 mL. There is no statistically significant difference in the MMO after the 6 months or equivalent (WMD: 9.62, 95% CI: 6.17 to 13.07, I^2^ = 99.1%, *p* = 0.392) follow-up between the two groups.

A sensitivity analysis is not performed as there is only one study with high risk of bias.

### 3.6. Pain (VAS)

The forest plots of the pooled WMD, between the <150 mL and ≥150 mL groups, in the reduction in VAS, are summarized in Figure 3. All included studies demonstrated a reduction in the VAS after arthrocentesis (range −1.23–−8.20), with a greater improvement in the VAS in the group with a lavage volume <150 mL. There is no statistically significant difference in the VAS after the 6 months or equivalent (WMD: −4.91, 95% CI: −3.89 to −5.93, I^2^ = 97.9%, *p* = 0.696) follow-up between the two groups.

A sensitivity analysis is not performed as there is only one study with a high risk of bias.

### 3.7. Assessment of the Publication Bias

The publication bias was assessed using the funnel plot techniques and Begg’s rank test. The funnel plots of the MMO and VAS mean differences were both not in a severe asymmetry, which are shown in Figure 4 and Figure 5. These implied the small study effect and thus a publication bias was not significant. Begg’s rank test also suggests no significant publication bias in the MMO (*p* = 0.163 in overall and 0.640 and 0.086 in the subgroups) and the VAS differences (*p* = 0.155 in overall and 0.210 and 0.734 in the subgroups).

## 4. Discussion

In the management of arthrogenous TMDs, arthrocentesis has become a standard treatment option, due to its high efficacy and safety [14,43]. However, different techniques of arthrocentesis exist in the literature and in clinical practice, such as additional injection materials into the superior joint space [41,42,44,45,46,47,48], the use of ultra-sound guidance [49,50,51,52] single versus double puncture techniques [30,32,53,54] and the timing of the procedure [19,55]. Specifically, the ideal irrigation volume for arthrocentesis of the TMJ remains a controversy. For example, in the studies included in this review, the irrigation volume ranged from 60 mL to 301 mL. Studies regarding the ideal irrigation volume for the TMJ arthrocentesis are few and far between, let alone the prospective clinical trials. If arthrocentesis performed with a smaller irrigation volume results in similar clinical outcomes, then arthrocentesis with a larger irrigation volume would be unnecessary, and the procedure could be completed in a timely fashion with increased patient comfort. Unfortunately, such a recommendation is difficult to make, due to the obvious knowledge gap at present. To the authors’ knowledge, the present study is the first systematic review and meta-analysis to investigate the ideal irrigation volume for the TMJ arthrocentesis.

The results of our study suggested that arthrocentesis with a smaller irrigation volume (<150 mL) may be superior to that performed with a larger irrigation volume (≥150 mL), in terms of pain reduction and jaw function. Our results showed that there was a greater improvement in the MMO and VAS in the group with a small irrigation volume (<150 mL). Nevertheless, the evidence may be inconclusive as there was no statistically significant difference between the smaller (<150 mL) and larger (≥150 mL) irrigation volumes in the MMO (WMD: 9.62, 95% CI: 6.17 to 218 13.07, I^2^ = 99.1%, *p* = 0.392) and VAS (WMD: −4.91, 95% CI: −3.89 to −5.93, I^2^ = 236 97.9%, *p* = 0.696). The pooled analyses represented by the forest plots are, however, from a limited number of studies. Although whether such difference seen in this study represents any actual clinical significance, is unknown, it may be safe to propose that arthrocentesis with a smaller volume (<150 mL), is at least as effective as that performed with a larger volume. However, since only the data from the 6-month follow-up was extracted from the included studies for the meta-analysis, it is not possible to say whether this is also true in the longer term.

The view that a smaller volume used for arthrocentesis of the TMJ may be just as effective as a larger volume from the current study, is shared with the few studies in the current literature on the topic. In a clinical study by Grossman et al., arthrocentesis of the TMJ was used to treat patients presented with disc displacement without reduction [22]. The patients were divided into two groups, with 50 mL or 200 mL of irrigation volume used in the procedure. While it was found that the favorable clinical outcomes were seen in both groups, in terms of pain reduction and improvement of the jaw function, no significant difference was found between the two groups. In another randomized controlled trial in 2017, no statistical significant difference was found between the groups with an irrigation volume of either 100 mL or 250 mL, while the clinical improvement of the TMD symptoms were seen in both groups [24]. Moreover, in a recent cadaveric study, it was found that 25 mL of the irrigation solution was sufficient to remove methylene blue from the TMJ space of fresh human cadavers [56]. In contrast, in a clinical trial by Kaneyama et al., it was suggested that the ideal lavage volume for the removal of inflammatory mediators from the TMJ joint space, was between 300–400 mL, although the statistical analysis of the clinical variables of the TMDs, such as pain reduction and mouth opening, were not reported in that study [57]. Therefore, more clinical studies are required in order to further understand the ideal irrigation volume for TMJ arthrocentesis.

There were a number of limitations to the current study. Due to the paucity of randomized controlled studies in the literature, it was not possible to compare the results of those studies performed with control groups that investigated the effect of different irrigation volumes. Rather, the results of the randomized controlled trials, which have reported the irrigation volume and outcomes of arthrocentesis and not focusing on the irrigation volumes, were pooled for our meta-analysis. This methodology is not ideal and therefore the results from this study cannot be interpreted with high certainty. Moreover, confounding factors were present across the studies, such as different diagnoses of TMDs, different materials used for the lavage and intra-articular injection, different techniques of arthrocentesis across centers and the difference in the mean age and sex distribution in the included studies. 

The association of ethnic background and prevalence of TMD were discussed in a number of studies [58,59,60]. It has been a less discussed factor and deserved more discussion in the context of the arthrocentesis outcome. Nevertheless, due to a limited number of studies specifying the patients’ ethnicity in the arthrocentesis treatment, a comprehensive systematic review becomes difficult. Other factors, such as age, duration of the symptoms and oral habits are believed to affect the prognosis of arthrocentesis [61]. Therefore, ethnicity can be investigated as an effect modifier in the arthrocentesis when a different lavage volume is applied. A more in depth understanding of the relationship between ethnicity, age, gender and different irrigation volumes of arthrocentesis is feasible when a diverse population is included in further studies. 

Moreover, only 6-months of data were selected for the meta-analysis, the effects of the different irrigation volume at different time points are unknown. Further review on the impact of the irrigation volume can be analyzed according to the duration of the follow-up. In fact, one study had identified the impact of the follow-up time of the different treatment for TMDs, by performing sub-group analyses, according to the duration of follow-up [14]. Therefore, more randomized controlled trials of arthrocentesis involving different follow-up periods are required before a final conclusion can be drawn. 

## 5. Conclusions

In conclusion, the current systematic review and meta-analysis suggests that arthrocentesis of the TMJ is at least as effective, if not more, when a smaller lavage volume is used (<150 mL). However, due to the limitations with the methodology and confounding factors, the evidence is weak at this time. Future randomized clinical trials are needed to better understand the clinical outcomes related to the different irrigation volumes used for the arthrocentesis of the TMJ.

## Figures and Tables

**Figure 1 diagnostics-12-02622-f001:**
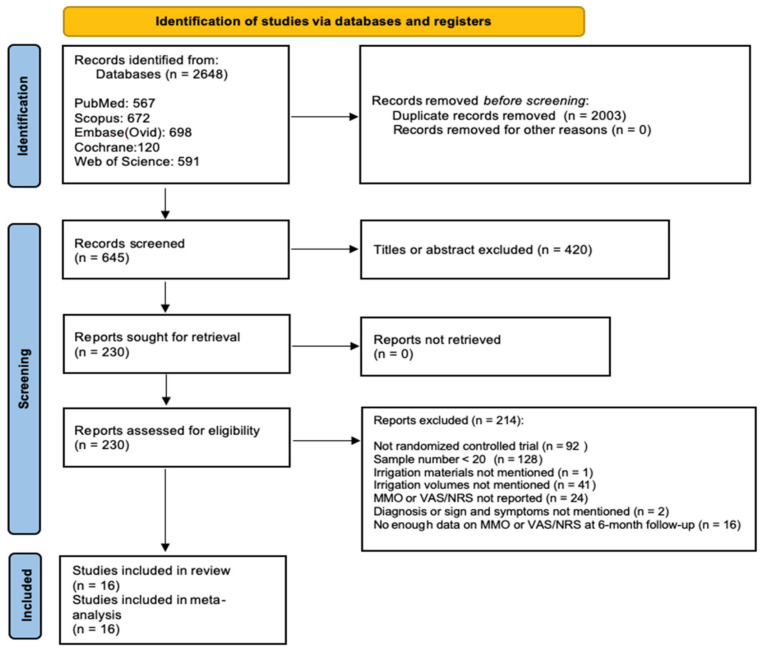
PRISMA 2020 flow diagram.

**Figure 2 diagnostics-12-02622-f002:**
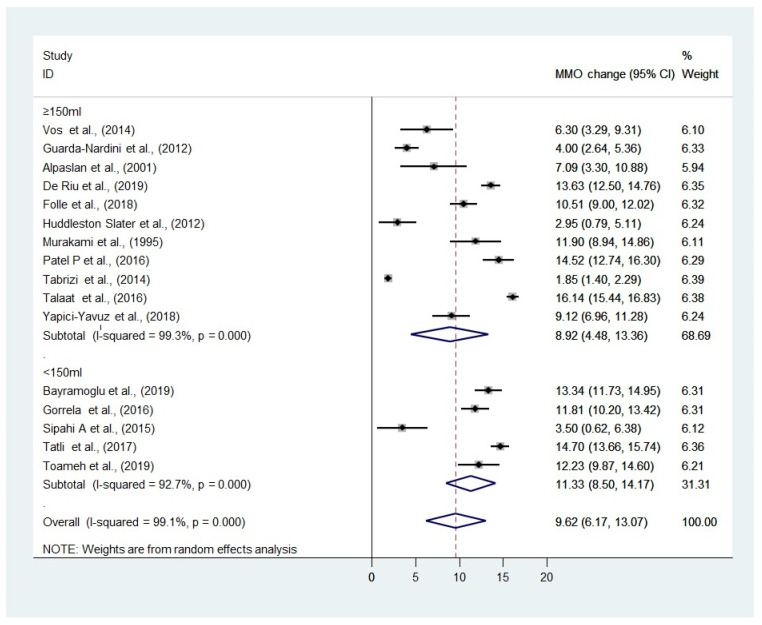
Forest plot using the random effects models showing the weighted mean difference (WMD) in the maximal mouth opening (MMO) between the two groups: all studies included. (Heterogeneity between the studies quantified using the I^2^ and *p*-value of the Chi^2^ test were listed) [19,23,29,30,31,32,33,34,35,36,37,38,39,40,41,42].

**Figure 3 diagnostics-12-02622-f003:**
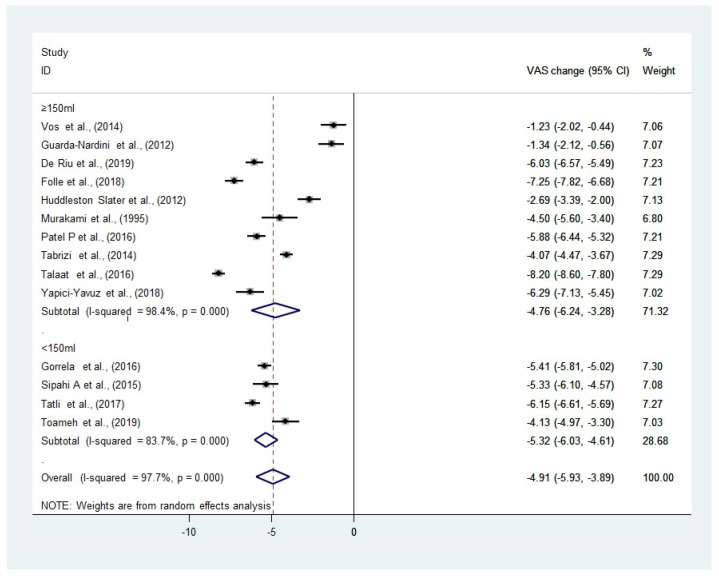
Forest plot using the random effects models showing the weighted mean difference (WMD) in the VAS between the two groups: all studies included. (Heterogeneity between the studies quantified using I^2^ and the *p*-value of the Chi^2^ test were listed) [19,23,31,32,33,34,35,36,37,38,39,40,41,42].

**Figure 4 diagnostics-12-02622-f004:**
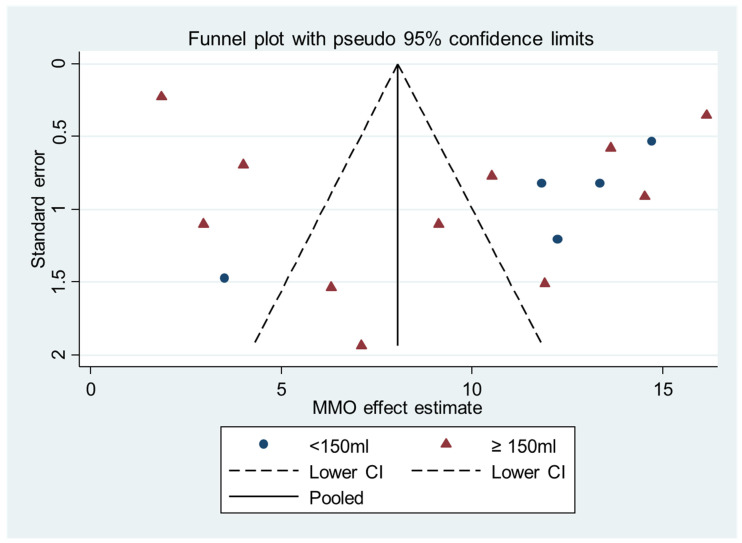
Funnel plot of the MMO mean differences.

**Figure 5 diagnostics-12-02622-f005:**
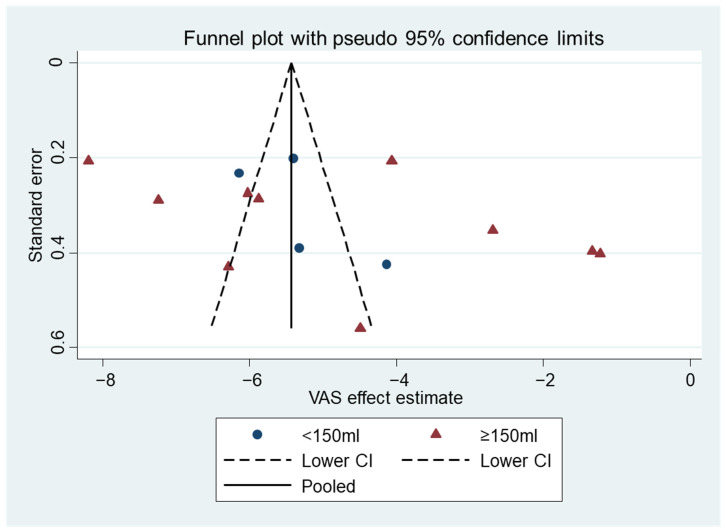
Funnel plot of the VAS mean differences.

**Table 1 diagnostics-12-02622-t001:** Details of the studies included.

Author	Year	Study Design	Subgroup	No. of Patients Who Underwent Arthrocentesis	Age at Treatment	Adjunctive Treatment	Irrigation Volume	(A: <150 mL, B: >150 mL)	Irrigation Material
Vos et al., (2014) [19]	2014	RCT	Arthrocentesis only	40	38.3		300 mL	B	NaCl
Guarda-Nardini et al., (2012) [23]	2012	RCT	SN grp	38	54.2		At least 300 + 1	B	Saline + HA
			TN grp (Control)	40	56.9			B	
Alpaslan et al., (2001) [29]	2001	RCT	Grp A (Control)	8	27		200–300	B	Saline
			Grp B	23			200–300 + 1	B	Saline + SH
Bayramoğlu et al., (2019) [30]	2019	RCT	SPA grp	16	25.9		100	A	Ringer’s lactate
			DPA grp (Control)	16	25.75			A	
De Riu et al., (2019) [31]	2019	RCT	HA grp (Control)	15	44.5		200–250 + 2	B	Ringer’s solution + SH
			BMNc grp	15	48.2		200–250 + 2	B	Ringer’s solution + BMNc (Bone marrow nucleated cell)
Folle et al., (2018) [32]	2018	RCT	SPA grp	13	37.38		300 + 1	B	Saline + SH
			DPA grp (Control)	13	30.77			B	
Gorrela et al., (2016) [33]	2016	RCT	A grp (Control)	31	Not provided	Post op PT	100	A	Saline
			A + SH grp	31			100 + 1	A	Saline + SH
Huddleston Slater et al., (2012) [34]	2012	RCT	Group 1 (Control)	14	33.9	1 cc saline (placebo)	300 mL	B	Saline
			Group 2 (+dexamethasone)	14	32.6	1 cc Dexamethasone		B	
Murakami et al., (1995) [35]	1995	RCT	Group II: arthrocentesis	20	31.2		Nitzan’s (2–3 mL Ringer’s + 200 mL lactated Ringer’s + 1 mL Celestone Soluspan)	B	Nitzan’s (2–3 mL Ringer’s + 200 mL lactated Ringer’s + 1 mL Celestone Soluspan)
Patel et al., (2016) [36]	2016	RCT	Grp 1: Arthrocentesis only (Control)	15	Mean age not reported: 21–30 (43.33%)			B	Ringer’s lactate
			Grp 2: Arthrocentesis + HA	15		Hyaluronic acid	2 mL (distend) + 200–300 mL	B	
Sipahi et al., (2015) [37]	2015	RCT	1 mL 5% Ringer’s lactate (Control)	10	Mean age not reported: (16–50)		60–100 mL	A	Ringer’s lactate
			Morphine 0.01 g made up to 10 mL Ringer’s lactate	10				A	
			Tramadol 50 mg mixed with 5% Ringer’s lactate 1 mL	10				A	
Tabrizi et al., (2014) [38]	2014	RCT	With Ringer only (Control)	30	28		2 mL saline (distend) + 200 mL	B	Ringer’s lactate
			With Ringer + dexamethasone	30	27.07	With 8 mg dexamethasone		B	
Talaat et al., (2016) [39]	2016	RCT	Single needle	28	26.025	1 mL HA	300 mL	B	Saline
			Double needle (Control)	28				B	
Tatli et al., (2017) [40]	2017	RCT	Arthrocentesis only (Control)	40	35.2	2 mL HA	120 mL	A	NaCl
			Arthrocentesis + Stabilization splint	40	38.9			A	
Toameh et al., (2019) [41]	2019	RCT	Arthrocentesis only (Control)	10	40.53		5 mL (distend) + 100 mL	A	Ringer’s lactate
			Arthrocentesis + HA	10	38.26	HA		A	
			Arthrocentesis + PRP	10	37.82	PRP		A	
Yapıcı-Yavuz et al., (2018) [42]	2018	RCT	Arthrocentesis + SH (diff in abstract and methods)	44	Not reported		Nitzan’s (2–3 mL Ringer’s + 200 mL lactated Ringer’s + 1 mL Celestone Soluspan)	B	Nitzan’s (2–3 mL Ringer’s + 200 mL lactated Ringer’s + 1 mL Celestone Soluspan)
			Arthrocentesis + methylprednisolone acetate					B	
			Arthrocentesis + tenoxicam					B	

**Table 2 diagnostics-12-02622-t002:** Risk of bias assessment.

Study ID	D1	D2	D3	D4	D5	Overall
Vos et al., (2014) [19]						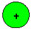
Guarda-Nardini et al., (2012) [23]						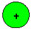
Alpaslan et al., (2001) [29]						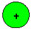
Bayramoğlu et al., (2019) [30]						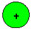
De Riu et al., (2019) [31]						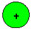
Folle et al., (2018) [32]						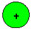
Gorrela et al., (2016) [33]						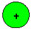
Huddleston Slater et al., (2012) [34]						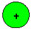
Murakami et al., (1995) [35]	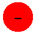	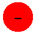	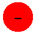			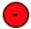
Patel et al., (2016) [36]						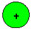
Sipahi et al., (2015) [37]						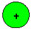
Tabrizi et al., (2014) [38]						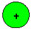
Talaat et al., (2016) [39]			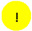			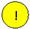
Tatli et al., (2017) [40]						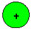
Toameh et al., (2019) [41]	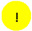					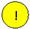
Yapıcı-Yavuz et al., (2018) [42]	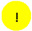					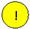


 Low risk; 
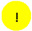
 Some concerns; 
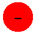
 High risk; D1: Randomisation process; D2: Deviations from the intended interventions; D3: Missing outcome data; D4: Measurement of the outcome; D5: Selection of the reported result.

## Data Availability

Data is available in the manuscript.

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
