# Peer review of "Lavage Volume of Arthrocentesis in the Management of Temporomandibular Disorders: A Systematic Review and Meta-Analysis"

_diagnostics, 2022, doi:10.3390/diagnostics12112622_

Round 1
Reviewer 1 Report
Dear Authors,
The aim of this study was to investigate the most effective lavage volume of arthrocentesis in the management of temporomandibular disorders. The results revealed the group with lavage volume <150ml have a greater improvement in mouth opening and pain reduction.
The study is of scientific interest and in line with the aims of the journal.
The author guidelines have been respected and the work is well written. The only concern was that the electronic search was updated to December 5, 2020.
Introduction
- It was useful to report that according to the Diagnostic Criteria for TMD (DC/TMD) Axis I, TMD can be divided in myogenic disorders (Group I) and intra-capsular disorders, including disc displacements (Group II) or arthralgia, arthritis, and arthrosis (Group III) (Shiffman 2014). Moreover, you should report a brief section reporting scientific literature of clinical conditions in which arthocentesis of TMJ could be useful (disc displacement for example, or only arthralgia.).
- “The first line approach involves non-surgical treatment that includes soft diet, pharmacotherapy such as non-steroidal anti-inflammatory drugs (NSAIDs), occlusal splint therapy, and physiotherapy.” It could be useful to clarify that conservative approaches could be useful especially in patients with both myogenic and arthrogenous temporomandibular disorders. Please discuss and cite “Ferrillo et al. Efficacy of rehabilitation on reducing pain in muscle-related temporomandibular disorders: A systematic review and meta-analysis of randomized controlled trials. J Back Musculoskelet Rehabil. 2022 Feb 18. doi: 10.3233/BMR-210236.”; “Andre A, Kang J, Dym H. Pharmacologic Treatment for Temporomandibular and Temporomandibular Joint Disorders. Oral Maxillofac Surg Clin North Am. 2022 Feb;34(1):49-59. doi: 10.1016/j.coms.2021.08.001”.; “Soni A. Arthrocentesis of Temporomandibular Joint- Bridging the Gap Between Non-Surgical and Surgical Treatment. Ann Maxillofac Surg. 2019 Jan-Jun;9(1):158-167. doi: 10.4103/ams.ams_160_17.”; “de Sire et al. Oxygen-Ozone Therapy for Reducing Pro-Inflammatory Cytokines Serum Levels in Musculoskeletal and Temporomandibular Disorders: A Comprehensive Review. Int J Mol Sci. 2022 Feb 25;23(5):2528. doi: 10.3390/ijms23052528”.
Material and methods
- Please modify the Figure 1 according to PRISMA 2020 flow diagram (Page MJ, McKenzie JE, Bossuyt PM, Boutron I, Hoffmann TC, Mulrow CD, et al. The PRISMA 2020 statement: an updated guideline for reporting systematic reviews. BMJ 2021;372:n71. doi: 10.1136/bmj.n71).
- For risk of bias assessment, please modify according to Rob2 (Sterne JAC, Savović J, Page MJ, Elbers RG, Blencowe NS, Boutron I, Cates CJ, Cheng HY, Corbett MS, Eldridge SM, Emberson JR, Hernán MA, Hopewell S, Hróbjartsson A, Junqueira DR, Jüni P, Kirkham JJ, Lasserson T, Li T, McAleenan A, Reeves BC, Shepperd S, Shrier I, Stewart LA, Tilling K, White IR, Whiting PF, Higgins JPT. RoB 2: a revised tool for assessing risk of bias in randomised trials. BMJ. 2019 Aug 28;366:l4898. doi: 10.1136/bmj.l4898.)
Reviewer 2 Report
This article is about the lavage volume of arthrocentesis in the management of temporomandibular disorders. Arthrocentesis has a high success rate and become a common therapeutic intervention for patients with TMDs. Therefore, the ideal lavage volume of arthrocentesis is an important standard that needs to be determined.
Although the paper is well-written and the goal of this study is relevant for field of research, some assumptions should be better explained:
(1) The 6 month follow up is mentioned several times in the text. Why was 6 months chosen as the criterion for follow-up? The 6 month follow up is mentioned several times in the text. Why was 6 months chosen as the criterion for follow-up?
(2) Have the authors considered the differences in the volume of lavage of arthrocentesis in different ethnic groups?
(3) The author's discussion of the data is a little lacking in the discussion section and could have gone into more depth on the data as well as the results.
Author Response
Please see the attachement.

Round 2
Reviewer 1 Report
Authors modified the text according to the suggestions.
In my opinion, it is suitable for publication.